# Effect of Dornase Alfa on the Lung Clearance Index in Children with Cystic Fibrosis: A Lesson from a Case Series

**DOI:** 10.3390/children9111625

**Published:** 2022-10-26

**Authors:** Vito Terlizzi, Giuseppe Fabio Parisi, Beatrice Ferrari, Chiara Castellani, Sara Manti, Salvatore Leonardi, Giovanni Taccetti

**Affiliations:** 1Cystic Fibrosis Regional Reference Center, Department of Paediatric Medicine, Meyer Children’s Hospital, 50139 Florence, Italy; 2Pediatric Respiratory and Cystic Fibrosis Unit, Department of Clinical and Experimental Medicine, San Marco Hospital, University of Catania, 95121 Catania, Italy; 3Rehabilitation Unit, Meyer Children’s Hospital, 50139 Florence, Italy; 4Pediatric Unit, Department of Human and Pediatric Pathology “Gaetano Barresi”, AOUP G. Martino, University of Messina, Via Consolare Valeria, 1, 98124 Messina, Italy

**Keywords:** dornase alfa, DNase, lung clearance index, lung function, children, cystic fibrosis

## Abstract

Background: Dornase alfa (DNase) is the only mucus-degrading agent that has proven efficacy in cystic fibrosis (CF). Few studies have evaluated the effects of DNase on the lung clearance index (LCI). We report the experience of two CF centers in which LCI monitoring was used to evaluate the efficacy of DNase therapy. Methods: This is a prospective and observational study, evaluating the effects of DNase therapy on LCI values in three CF children followed at CF centers in Florence and Catania, Italy. In both centers, LCI was performed routinely, every 3–6 months, based on the clinical picture and severity of the lung disease. In this study, we evaluated the LCI before and after long-term DNase therapy. Results: DNase improved LCI values in the absence of respiratory exacerbations: in case n. 1 LCI decreased by 39% in 16 months (from 11.1 to 6.8); in case n. 2 by 20% in 12 months (from 9.3 to 7.4); in case n. 3 by 24% in 16 months (from 9.3 to 7.0). Conclusions: This case series confirms the efficacy of DNase therapy in CF children, as demonstrated by the LCI reduction in treated patients. Furthermore, our results suggest that LCI is a sensitive marker of disease and can be used for the evaluation of response to treatment.

## 1. Introduction

Cystic fibrosis (CF) is the most common inherited disease in the Caucasian population and it is caused by pathogenetic variants in a gene that encodes the cystic fibrosis transmembrane conductance regulator (CFTR) protein and it is expressed in many epithelial and blood cells. CF is characterized by progressive damage to the small airways, which over time, causes a reduction in lung function and the development of bronchiectasis due to triggering chronic inflammation mechanisms. This progression is objectively monitored through common lung function tests and imaging methods [1].

Conventional spirometry has long been considered the primary test of respiratory function deficits in children and adults. However, recent evidence suggests that conventional spirometry is not sensitive for the early detection of lung damage affecting the small airways or the evaluation of homogeneity of air ventilation [2,3].

For these reasons, alternative techniques have been implemented over the last decades. Multiple-breath washout (MBW) of inert tracer gases is known to be a noninvasive, sensitive, and feasible measure of lung function to detect peripheral airway changes in childhood [4,5,6,7]. The main index obtained from MBW tests is the lung clearance index (LCI), which measures the lung turnover necessary to reduce the concentration of tracer gas present at the beginning of the exam to 1/40 [8].

Previous cross-sectional studies in school-age children with CF have suggested that LCI is more sensitive than Forced Expiratory Volume in one second (FEV1) for detecting lung disease. Furthermore, it can detect mild lung-function abnormalities [9,10]. Finally, it was used in pediatric clinical trials with CFTR modulators as the main outcome measure [11,12,13].

A percentage change of greater than 15% in the LCI in preschool children can be considered physiologically relevant and greater than the biological variability of the test [14]. In the same way, an increase in LCI > 17% compared to previous LCI-measurement in clinically stable CF patients may indicate early lung disease progression [15].

Dornase alfa (DNase) is the only mucus-degrading agent that has proven to be efficacious in CF [16]. It reduces mucus viscosity in the lungs, promoting improved clearance of secretions. In this way DNase is associated with an improvement in lung function, a reduction in the number of pulmonary exacerbations in patients with CF [16,17]. Few studies have evaluated the effects of DNase on LCI in children with CF.

We report the experience of two CF centers in which LCI monitoring was used to evaluate the efficacy of DNase therapy.

## 2. Materials and Methods

This was a prospective and observational study that evaluated the effects of DNase therapy on LCI values in three CF children followed at the CF centers of Florence and Catania, Italy. The study was conducted according to the principles of the Declaration of Helsinki. Informed consent was obtained from the children’s parents after the parents received a complete description of the study’s aims.

In both centers, LCI was performed routinely, every 3–6 months, based on the clinical picture and severity of the lung disease. For this study, we evaluated the LCI before and after long-term DNase therapy and correlated them with FEV1 values that were calculated according to the Global Lung Function Initiative [18].

Exhalyzer-D (EcoMedics AG, Duernten, Switzerland, software version 3.2.0) was used to conduct the LCI tests. The LCI upper limit of normal (ULN) was defined as 7.91 according to Anagnostopoulou et al. [19] for school-aged children.

Two physiotherapists conducted each LCI test, which was performed when the patients were in stable condition according to the American Thoracic Society (ATS) and the European Respiratory Society (ERS) consensus statement [20]. Washout repeats were excluded if there was evidence of a leak or a large difference between the LCI and functional residual capacity (FRC) measurements (>25% from the median). On the other hand, the session was approved if at least two or more technically acceptable trials were obtained [20].

A stable condition was defined as having no symptoms or signs of pulmonary exacerbation according to the CF Foundation criteria [21]. A children’s entertainment video was played as a distraction during testing. Device calibration and the spirometer were calibrated on each test day.

Pancreatic insufficiency (PI) was defined based on at least two values of fecal pancreatic elastase lower than 200 µ/g measured in the absence of acute gastrointestinal diseases [22,23].

*Pseudomonas aeruginosa* (Pa), *methicillin-sensitive Staphylococcus aureus* (MSSA), and *methicillin-resistant Staphylococcus aureus* (MRSA) chronic infection was defined using the modified Leeds criteria [24]. Pulmonary exacerbations were defined according to the CF Foundation’s criteria [21].

## 3. Results

### 3.1. Case #1

We report the case of a 13-year-old Caucasian child diagnosed with CF and PI based on positive newborn screening (NBS) [25] (immunoreactive trypsinogen 103 ng/mL, CFTR genotype: R1162X/R553X, sweat chloride: 105–106 mEq/L) and followed at the CF Center of Florence, Italy. In 2021, the clinical picture was characterized by a body mass index (BMI) of 17.0 kg/m^2^ (30th centile for age), a chronic obstructive pulmonary disease with FEV1 in the range of 80% to 84%, one exacerbation per year requiring oral antibiotics and chronic infection caused by MRSA. Positive expiratory pressure (PEP) mask respiratory physiotherapy was prescribed twice per day; no other respiratory therapies were indicated due to the absence of bronchiectasis, as seen on chest computed tomography (CT) scans and the low number of pulmonary exacerbations (1–2 per year). At the age of 12 years, DNase therapy was prescribed based on a pathological LCI value (11.1), and a worsening of spirometry parameters (FEV1 75%). The subsequent LCI values resulted in marked improvement (>15% [14]) already after 3 months of therapy with a return to normal limits after 16 months. In addition, the FEV1 value also improved in the absence of respiratory exacerbations (Table 1). No other therapeutic changes were made, and no increase in sporting activities or variations in compliance with treatment were noted.

### 3.2. Case #2

Case #2 involved a 7-year-old child during a follow-up at the CF center in Catania, Italy, who had been diagnosed with CF and PI by positive NBS (blood immunoreactive trypsinogen 125 ng/mL, CFTR genotype: N1303K/F508del, sweat chloride: 109–110 mEq/L). PEP mask respiratory physiotherapy was prescribed twice per day. Furthermore, during the follow-up, the patient developed chronic Pa colonization and received therapy with inhaled tobramycin and colistin every other month. He had a low number of pulmonary exacerbations (1 per year) requiring i.v. antibiotics. DNase therapy was started at the age of 7 years and 6 months following the finding of bronchiectasis on a chest CT scan, which was associated with a FEV1 of 101% and a pathological LCI (9.3). His BMI was 17.1 kg/m^2^ (75th centile for age). After four months, we recorded a significant reduction in the LCI value up to 7.4 after one year of therapy (Table 2). Regarding conventional spirometry, the values always remained within the normal range (Table 2). Furthermore, no pulmonary exacerbation was reported.

### 3.3. Case #3

We report the case of an 11-year-old Caucasian child diagnosed with CF and PI based on positive NBS [25] (blood immunoreactive trypsinogen 105 ng/mL, CFTR genotype: F508del/L1065P, sweat chloride: 112–114 mEq/L), followed at the CF Center of Florence, Italy. Before DNase therapy, the clinical picture was characterized by a BMI of 20.54 kg/m^2^ (90th centile for age), a chronic obstructive pulmonary disease with FEV1 in the range of 80% to 85%, and chronic infection caused by MSSA. He had a low number of pulmonary exacerbations (1–2 per year). PEP mask respiratory physiotherapy was prescribed twice per day. DNase therapy was started at the age of 11 years following the finding of bronchiectasis on a chest CT scan, which was associated with an FEV1 of 80% and a pathological LCI (9.31).

The subsequent LCI values resulted in a significant improvement already after six months of therapy. An insignificant increase (<15%) was noted after 12 months. In addition, the FEV1 value was also improved in the absence of respiratory exacerbations (Table 3). No other therapeutic changes were made, and no variations in compliance with treatment were noted. Furthermore, according to the Italian legislative directives [26], the patient started elexacaftor/tezacaftor/ivacaftor at 12 years and continued regular DNase therapy. The combination of the two drugs resulted in a further reduction in the LCI up to a value below the ULN for age (7.91) [19].

## 4. Discussion

In this study, we report the positive effects of DNase on LCI values (and so on the inhomogeneity of global pulmonary ventilation) in three children with CF. Our data confirm that CF children with a pathological LCI value can benefit from therapy with DNase.

In the first case, DNase therapy was prescribed based only on the finding of pathological LCI values, associated with a worsening of spirometric parameters. The patient showed no bronchiectasis at the chest CT scan and a reduced number of respiratory exacerbations. The therapy resulted in an improvement in both FEV1 and LCI, and signs of improvements in ventilation in the airways following DNase mucolytic action.

In the second case, we show a pathological LCI value in the presence of normal FEV1; this finding is typical of the early pulmonary disease of CF patients as it reflects a small airway involvement in the formation of bronchiectasis, without alterations in the dynamic lung volumes. Following DNase therapy, decreased LCI values were demonstrated, indicating improved small airway ventilation.

The third case further confirms the efficacy of DNase in the reduction in ventilatory inhomogeneity and also in preventing pulmonary exacerbations. Furthermore, the effect seems to become even greater when the triple combination CFTR modulator therapy is started.

MBW is a technique used to assess the quality of ventilation and is a valid option for covering the lack of surveillance in early disease. Over the last several years, the use of MBW has been revived since it is non-invasive, requires minimal cooperation from the patient, and small airways are its target. LCI is the measure obtained by using the MBW test. More widespread use of LCI among CF centers seems to be occurring [27,28,29]. LCI capabilities have been evaluated from different viewpoints, such as its capacity to detect and track early lung disease or to indicate any disease progression. Ramsey et al. confirmed the relationship between a high LCI value in school-aged children and lung disease based on CT findings. The authors demonstrated in pre-school-aged children that LCI correlated with total disease extent, and in school-aged ones, LCI was an index of bronchiectasis and reflected totally extended disease. In these two groups, LCI showed a good predictive positive value of 83% to 86%, while the predictive negative value was 50% to 55% in bronchiectasis detection [30]. Although LCI is sensitive to bronchiectasis, CT is still mandatory to identify bronchiectasis in CF patients after which LCI might be a surrogate to use during surveillance of an already assessed situation. In a study by Belessis et al., the reason for high LCI values in 21% of infants, despite antibiotic treatment, was clarified. Infants were studied using LCI and bronchoalveolar lavage (BAL). LCI seems to be associated to lung inflammation (as shown by elevated interleukin-8 and neutrophilic enzymes and cells) [31]. The difference in LCI values between CF infants with and without infection was not statistically significant. However, LCI correlated with the pathogen load. Infants infected with Pa had higher LCI values (mean 7.92 ± 1.16) than children with no bacterial colonization (7.02 ± 0.56). Furthermore, Pa infection is associated with an increase in inflammatory markers in BAL samples with respect to children infected with other pathogens because this microorganism leads to an intense neutrophil response [31].

These data may explain the elevated LCI values with normal FEV1 in case number 2.

Regarding LCI as an outcome measure in clinical trials, several studies clarify its use. Clinical responses to Ivacaftor, an orally CFTR potentiator, were tested in children with preserved lung function (FEV1 > 90%) and evaluated using LCI as an endpoint given that lung function was normal. The study results indicated an improvement, which was represented as an LCI reduction in patients treated with Ivacaftor [32]. In 2017, the first placebo-controlled phase 3 trial using LCI as the primary endpoint was performed with a 1-unit LCI decline as the objective. The efficacy and safety of Lumacaftor–Ivacaftor were tested in children with FEV1 > 70% [33]. The choice of LCI as the endpoint in clinical trials is more advantageous because a small size sample can be enough to achieve acceptable results compared to a sample studied using FEV1 as the endpoint [34]. Thus, LCI seems to be one of the useful tools in early/mild disease surveillance, and this value will be probably recognized by international guidelines as a standard in conjunction with spirometry to track CF disease in clinical centers.

Regarding LCI as a tool to evaluate other inhaled drugs, Ratjen et al. demonstrated the efficacy of early treatment with hypertonic saline in children aged 3–6 years as evaluated by change in the LCI measured by nitrogen MBW from baseline to week 48 [35].

The effects of DNase on LCI have been poorly evaluated in the literature. In 2011, Amin et al. tested DNase on 17 CF patients aged 6 to 18 years with FEV1 ≥ 80%. The drug was taken for four weeks followed by a washout period of four weeks. They detected significant improvements in LCI in the treated versus the placebo group (0.90 ± 1.44; *p* = 0.022). No significant differences in the value of FEV1 were noted, probably due to the good lung function of the selected patient group [36].

Lately, a single-center, randomized, controlled, parallel-group study evaluated the effects of a one-month withdrawal of nebulized DNase in 5–18-year-old children with CF. At the end of the suspension period, an increase in the LCI value (1.74 (95% confidence interval: 0.62; 2.86)) and a decrease in the FEV1 value (−6.8% predicted) were observed [37].

Our case differs from the previous ones: first of all, the length of the observation period (1 month in the Amin study vs. 12–16 months of our patients) demonstrates, in an original way, how the effects of the drug are prolonged and ameliorative over time [36]; unlike Voldby, however, we did not have to suspend the treatment to check for any worsening which, moreover, would have been unethical [37].

The rationale for why DNase would lead to an improvement in LCI lies in its mechanism of action. Recombinant human DNase is a genetically engineered analog of the natural human enzyme that is capable of inducing the fragmentation of extracellular DNA. Retention of purulent viscous airway secretions contributes to decreased lung function and exacerbates infections. Purulent secretions contain very high concentrations of extracellular DNA, a viscous polyanion released in the degeneration process of leukocytes that accumulate in response to infection. In vitro, DNase hydrolyzes DNA in the sputum of patients with CF and leads to a significant reduction in its viscoelasticity [38,39]. This action, associated with respiratory physiotherapy, causes a reduction in airway obstruction caused by mucus and leads to an improvement in ventilatory homogeneity.

The most recent Cochrane review published on the efficacy of DNase CF showed that this therapy improves lung function (measured as FEV1%) over a period of one month to 2 years [16]. The best results appear to be seen in patients with the moderate and non-severe disease. However, when compared with the other mucolytic agents used in CF (mannitol and hypertonic saline) it was not possible to demonstrate greater efficacy than the other treatments [16,40]. These results were also confirmed in the few studies that only considered children [16,40].

## 5. Conclusions

This case series confirms the efficacy of DNase therapy in CF children, as demonstrated by the reduction in LCI in treated patients, even for an observation period of up to 16 months. Furthermore, it suggests that LCI is a sensitive marker of disease and can be used for the evaluation of response to treatment. DNase therapy should be early evaluated in CF children with pathological LCI values, even in the absence of bronchiectasis at chest CT scan. The major limitation is represented by the small sample of patients (n. 3).

## Figures and Tables

**Table 1 children-09-01625-t001:** Time course of LCI values at time 0 and after dornase alfa therapy.

	LCI 2.5	Percentage Change Compared to the First Test	FEV1 (% of Predicted)
T0 (13 years)	11.1		75
After 3 months	7.75	−31%	84
After 9 months	7.93	−29%	90
After 16 months	6.87	−39%	96

Abbreviations: LCI: lung clearance index; FEV1: Forced Expiratory Volume in the first second.

**Table 2 children-09-01625-t002:** Time course of LCI values at time 0 and after dornase alfa therapy.

	LCI 2.5	Percentage Change Compared to the First Test	FEV1 (% of Predicted)
T0 (7 years)	9.3		101
After 4 months	7.8	−16%	103
After 8 months	7.5	−19%	105
After 12 months	7.4	−20%	106

Abbreviations: LCI: lung clearance index; FEV1: Forced Expiratory Volume in the first second.

**Table 3 children-09-01625-t003:** Time course of LCI values at time 0 and after dornase alfa therapy.

	LCI 2.5	Percentage Change Compared to the First Test	FEV1 (% of Predicted)
T0 (11 years)	9.31		80
After 6 months	7.90	−15%	89
After 12 months	8.19	−12%	96
After 16 months *	7.06	−24%	92

* Concomitant therapy with elexacaftor/ivacaftor/tezacaftor for three months. Abbreviations: LCI: lung clearance index; FEV1: Forced Expiratory Volume in the first second.

## Data Availability

Original data are available upon request to the authors.

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
