# Peer review of "Effect of Dornase Alfa on the Lung Clearance Index in Children with Cystic Fibrosis: A Lesson from a Case Series"

_children, 2022, doi:10.3390/children9111625_

Round 1

Reviewer 1 Report

Terlizzi et al. reported a case series of 3 children to illustrate the usefulness of LCI to monitor DNase efficacy. This is a nice, but small (considering CF in children is not a so-rare condition), case series, well documented with interesting discussion about use of LCI.

Major comments:

- Some parts of the manuscript if really close to a previous recent publication of the same group (Terlizzi et al. Italian Journal of Pediatrics 2022). I assume that it is not so easy to re-write differently same information, but re-check to avoid any copy-past.

- The conclusion of this case series is obviously too affirmative. It shows a nice enthusiasm of the authors, but unfortunately, a case series of 3 patients will never be enough to confirm the efficacy of a drug. Otherwise, there would be no need for clinical trials! The authors should redefine what their case series add to previous studies focused on DNase efficacy assessed by LCI in CF children (Amin et al. 2010 and Voldby et al. 2021), considering these are randomized clinical trials.

- The manuscript mainly focus on MBW, well describing the usefulness of LCI measurement in the introduction, as well as the applications, limitations and main results in CF in the discussion. However, the authors should better detaille the DNase use and give insight on efficacy (which populations/disease studied, conflicting results in children with CF especially when compared with alternative medications, which parameters already used to assess efficacy, etc…). For sure, this would increase the interest in the use of LCI to monitor DNase efficacy in CF children, as the authors did, even if the authors give part of those information at the end of the discussion.

Minor comments

- Abstract line 29 and M&M line 82: “before and after DNase therapy” is a little bit confusing, as we do not know if the authors measured LCI before and after a single DNase nebulisation, or in the months before and after the introduction of DNase as a new long-term therapy. I suggest to clarify, for example in writing “before and after introduction of a long term DNase therapy”

- Abstract line 35: “in small airways” looks inappropriate… The authors discussed the pharmacodynamics of DNase and the impact on small airways disease in CF in the manuscript, but they did not properly study the effect on small airways. Therefore, this study did not add data on DNase pharmacodynamics and the authors cannot conclude that this case series demonstrate effect of DNase on small airways. I suggest replacing with “in CF children”.

- Introduction line 43: use “variant” or “pathogenic variant” instead of “mutation”

- Introduction line 47: ”progression” would be better than “evolution”

- Introduction line 47: I suggest adding “objectively” (…is objectively monitored…)

- Introduction line 53: “”last few years”… MBW is applied to CF patients since more than 15 years. Please modify.

- Result lines 109, 129 and 142 : “CFTR” should be in italic when referring to the gene

- Results: The number of exacerbations is a major clinical endpoint to assess the efficacy of DNase (as stated in the introduction by the authors), but is not detailed in the case reports except for the third one. Please add information about number of exacerbation before and after DNase in case 1 and 2.

- Discussion line 182: it’s not true that MBW doesn’t required cooperation, switch to “reduced cooperation” or at least “minimal cooperation”

Author Response

Terlizzi et al. reported a case series of 3 children to illustrate the usefulness of LCI to monitor DNase efficacy. This is a nice, but small (considering CF in children is not a so-rare condition), case series, well documented with interesting discussion about use of LCI.

Answer: Dear reviewer, thank you very much for your comments. I hope to provide the clarifications needed to improve the article.

Major comments:

- Some parts of the manuscript if really close to a previous recent publication of the same group (Terlizzi et al. Italian Journal of Pediatrics 2022). I assume that it is not so easy to re-write differently same information, but re-check to avoid any copy-past.

Answer: Dear reviewer, we modified some paragraphs to avoid repetitions.

- The conclusion of this case series is obviously too affirmative. It shows a nice enthusiasm of the authors, but unfortunately, a case series of 3 patients will never be enough to confirm the efficacy of a drug. Otherwise, there would be no need for clinical trials! The authors should redefine what their case series add to previous studies focused on DNase efficacy assessed by LCI in CF children (Amin et al. 2010 and Voldby et al. 2021), considering these are randomized clinical trials.

Answer: We added the main limitation of this study in the conclusion section. Moreover, we highlighted the differences between our study and that of Amin and Voldby in the discussion.

 - The manuscript mainly focus on MBW, well describing the usefulness of LCI measurement in the introduction, as well as the applications, limitations and main results in CF in the discussion. However, the authors should better detaille the DNase use and give insight on efficacy (which populations/disease studied, conflicting results in children with CF especially when compared with alternative medications, which parameters already used to assess efficacy, etc…). For sure, this would increase the interest in the use of LCI to monitor DNase efficacy in CF children, as the authors did, even if the authors give part of those information at the end of the discussion.

Answer: We added some more information about DNase in the discussion.

Minor comments

- Abstract line 29 and M&M line 82: “before and after DNase therapy” is a little bit confusing, as we do not know if the authors measured LCI before and after a single DNase nebulisation, or in the months before and after the introduction of DNase as a new long-term therapy. I suggest to clarify, for example in writing “before and after introduction of a long term DNase therapy”

Answer: We clarified the sentence.

- Abstract line 35: “in small airways” looks inappropriate… The authors discussed the pharmacodynamics of DNase and the impact on small airways disease in CF in the manuscript, but they did not properly study the effect on small airways. Therefore, this study did not add data on DNase pharmacodynamics and the authors cannot conclude that this case series demonstrate effect of DNase on small airways. I suggest replacing with “in CF children”.

Answer: Thank you. As you suggested, we modified the sentence. 

- Introduction line 43: use “variant” or “pathogenic variant” instead of “mutation”

Answer: As you suggested, we corrected the mistake.. 

- Introduction line 47: ”progression” would be better than “evolution”

Answer: As you suggested, we changed the text.  

- Introduction line 47: I suggest adding “objectively” (…is objectively monitored…)

Answer: As you suggested, we changed the text.  

- Introduction line 53: “”last few years”… MBW is applied to CF patients since more than 15 years. Please modify.

Answer: As you suggested, we changed the text.  

- Result lines 109, 129 and 142 : “CFTR” should be in italic when referring to the gene

Answer: As you suggested, we changed the text.  

- Results: The number of exacerbations is a major clinical endpoint to assess the efficacy of DNase (as stated in the introduction by the authors), but is not detailed in the case reports except for the third one. Please add information about number of exacerbation before and after DNase in case 1 and 2.

Answer: As you suggested, we added some details about pulmonary exacerbations before and after treatment.  

- Discussion line 182: it’s not true that MBW doesn’t required cooperation, switch to “reduced cooperation” or at least “minimal cooperation”

Answer: Thank you. As you suggested, we changed the text.  

Reviewer 2 Report

This prospective and observational study evaluates the effects of DNase therapy on LCI values in three CF children followed at CF centers in Florence and Catania, Italy. In both centers, LCI was performed routinely, every 3–6 months, based on the clinical picture and severity of the lung disease. In this study, the authors evaluated the LCI before and after DNase therapy. Results DNase improved LCI values in the absence of respiratory exacerbations: in case n. 1, LCI decreased by 39% in 16 months (from 11.1 to 6.8); in case n. 2, by 20% in 12 months (from 9.3 to 7.4); in case n.3 by 24% in 16 months (from 9.3 to 7.0). This case series confirms the efficacy of DNase therapy in small airways, as demonstrated by the LCI reduction in treated patients.

 The manuscript is of broad and cross-disciplinary interest. The topic and content discussed in this manuscript are within the journal's scope. The organization and subsections are also appropriate. The manuscript is structured and presented in a reader-friendly manner.

Author Response

Answer: Dear reviewer, thank you for your positive comments. 

Round 2

Reviewer 1 Report

The authors adequatly modified their manuscript.

At the end I still believe that the authors can not state that this study, incuding 3 case reports, "confirms the efficacy of DNase therapy". This conclusion is too affirmative and not support by the results. Please modify abstract conclusion, first paragraph of discussion and final conclusion.

Author Response

Dear reviewer, thank you.

We modified the abstract conclusions, the first paragraph of the discussion and the conclusions.